# Homo-PROTACs: bivalent small-molecule dimerizers of the VHL E3 ubiquitin ligase to induce self-degradation

Chiara Maniaci[1,2], Scott J. Hughes[1], Andrea Testa [1], Wenzhang Chen[1], Douglas J. Lamont[1], Sonia Rocha [3], Dario R. Alessi [2], Roberto Romeo[4] & Alessio Ciulli [1]

E3 ubiquitin ligases are key enzymes within the ubiquitin proteasome system which catalyze the ubiquitination of proteins, targeting them for proteasomal degradation. E3 ligases are gaining importance as targets to small molecules, both for direct inhibition and to be hijacked to induce the degradation of non-native neo-substrates using bivalent compounds known as PROTACs (for 'proteolysis-targeting chimeras'). We describe Homo-PROTACs as an approach to dimerize an E3 ligase to trigger its suicide-type chemical knockdown inside cells. We provide proof-of-concept of Homo-PROTACs using diverse molecules composed of two instances of a ligand for the von Hippel-Lindau (VHL) E3 ligase. The most active compound, CM11, dimerizes VHL with high avidity in vitro and induces potent, rapid and proteasome-dependent self-degradation of VHL in different cell lines, in a highly isoform-selective fashion and without triggering a hypoxic response. This approach offers a novel chemical probe for selective VHL knockdown, and demonstrates the potential for a new modality of chemical intervention on E3 ligases.

[1] Division of Biological Chemistry and Drug Discovery, School of Life Sciences, University of Dundee, James Black Centre, Dow Street, Dundee, Scotland DD1 5EH, UK. [2] Medical Research Council Protein Phosphorylation and Ubiquitylation Unit, School of Life Sciences, University of Dundee, James Black Centre, Dow Street, Dundee, Scotland DD1 5EH, UK. [3] Centre for Gene Regulation and Expression, School of Life Sciences, University of Dundee, James Black Centre, Dow Street, Dundee, DD1 5EH Scotland, UK. [4] Dipartimento di Scienze Chimiche, Biologiche, Farmaceutiche ed Ambientali, University of Messina, Polo Universitario Viale SS. Annunziata SNC, Messina, 98168, Italy. Correspondence and requests for materials should be addressed to A.C. (email: a.ciulli@dundee.ac.uk)

E3 ubiquitin ligases are emerging as attractive targets for small-molecule modulation and drug discovery[1–3]. E3s bring a substrate protein and ubiquitin in close proximity to each other to catalyze the transfer of a ubiquitin molecule to the substrate[4, 5]. Substrate ubiquitination can trigger different cellular outcomes[6], of which one of the best characterized is poly-ubiquitination and subsequent proteasomal degradation[7, 8]. The human genome comprises > 600 predicted E3 ligases that play important roles in normal cellular physiology and disease states, making them attractive targets for inhibitor discovery[9]. However, E3 ligases do not comprise deep and 'druggable' active sites for binding to small molecules[2]. Blockade of E3 ligase activity therefore requires targeting of protein–protein interactions (PPIs), and the often extended, flat and solvent-exposed PPI surfaces make it a challenge for drug design[10]. Only few potent inhibitors have been developed to date, mostly compounds that bind to the E3 substrate recognition site[2, 11], for example MDM2[12, 13], inhibitor of apoptosis proteins (IAPs)[14–16], the von Hippel-Lindau (VHL) ligase[17–19] and KEAP1[20, 21]. Inhibitors of E3:substrate interaction can exhibit a discrepancy in effective concentrations between biophysical binding and cellular efficacy[19, 22], due to competition from high-affinity endogenous substrates that markedly increase their cellular concentration as a consequence of the inhibition. This poses limitations, such as incomplete blockade of enzyme activity and the need to use high inhibitor concentrations, which can lead to off-target effects and cytotoxicity. Moreover, E3 ligases are multi-domain and multi-subunit enzymes, and targeting an individual binding site leaves other scaffolding regions untouched and other interactions functional. As a result, E3 ligase inhibition may be ineffective or fail to recapitulate genetic knockout or knockdown. New chemical modalities to target E3 ligases are therefore demanded.

E3 ligases are not merely targets for inhibition. Compounds of natural or synthetic origin have been discovered that bind to E3 ligases and promote target recruitment. These interfacial compounds induce de novo formation of ligase-target PPIs, effectively hijacking E3 activity towards the neo-substrates, for targeted protein degradation[23, 24]. One class of hijackers of E3 ligase activity comprises monovalent compounds. These so-called 'molecular glues' include the plant hormone auxin, which binds to the Cullin RING ligase (CRL) CRL1-TIR1 to target transcriptional repressor proteins of the Aux/IAA family;[25] the immunomodulatory drugs (IMiDs) thalidomide, lenalidomide, pomalidomide and analog CC-885, that all share binding to cereblon (CRBN), a subunit of the CRL4-CRBN ligase, and redirect CRBN activity to different substrates[26–31]. More recently, sulfonamide anti-cancer drug indisulam was found to induce degradation of the splicing factor RBM39 via recruiting CRL4-DCAF15 activity[32, 33]. Another class of degrader compounds that display a similar mechanism of action comprises bivalent molecules known as Proteolysis-Targeting Chimeras (PROTACs). PROTACs comprise two warheads—one for ligase recruitment and a second one for target-binding—joined by a linker[34]. Formation of a ternary complex between the PROTAC, the ligase and the target triggers proximity-induced target ubiquitination and degradation. Potent and cell-active PROTACs have been developed for recruiting different ligases, including CRL2-VHL[35–38], CRL4-CRBN[39–42], and IAPs[43, 44]. Targets successfully degraded by PROTACs include BET proteins Brd2, Brd3 and Brd4[35, 37–40], FKBP[39], protein kinases[36, 41], amongst others[36, 43]. An attractive feature of bivalent degrader molecules is their sub-stoichiometric catalytic activity[36], which does not require full occupancy of the target-binding site as with conventional inhibitors, leading to degrading concentrations that can be orders of magnitude lower than the inhibitory concentrations of their constitutive parts alone. Furthermore, induced target depletion can have a more sustained cellular effect compared to target inhibition, and can overcome compensatory cellular feedback mechanisms, such as increase in target levels[45]. Crucially, work from us and others have shown that PROTAC molecules can exhibit an added layer of selectivity for protein degradation beyond the intrinsic binding selectivity of the warhead ligand[35, 38, 41]. Our recent structural work with Brd4-selective PROTACs targeting CRL2-VHL revealed the importance of specific ligand-induced PPIs between the ligase and the target, which contribute to cooperative formation of stable and highly populated ternary complexes[38].

We hypothesized that it could be possible to trigger an E3 ligase to induce its own degradation, by designing tailored homo-bivalent PROTACs that recruit two molecules of the same E3 ligase. The idea was simple, namely that this compound class could act as chemical inducers of dimerization (CID)[46–48], forming a ternary complex in which the E3 acts as the enzyme and the neo-substrate at the same time. To provide proof-of-concept for the approach, we designed, synthesized and tested homo-bivalent molecules aiming to target CRL2-VHL, the E3 ligase that targets for ubiquitination and degradation the hypoxia inducible factor alpha subunit (HIF-α) under normoxic conditions[49]. We show the most active compound, CM11, made of two instances of a potent hydroxyproline (Hyp) containing VHL ligand[19], avidly forms a 1:2 complex with VHL, and induces potent and preferential isoform-selective degradation of VHL. We call these Homo-PROTACs, as a new modality to induce chemical knockdown of E3 ligases.

## Results

**Rational design**. Design of VHL Homo-PROTACs began with careful consideration of the position of derivatization on two potent VHL ligands recently characterized by our group, VH032 and VH298 (Fig. 1a, b)[18, 19]. To retain the strong binding affinity that characterizes the ligand, co-crystal structures were analyzed to identify solvent-exposed regions from where the ligands could be derivatized without perturbing their binding modes (Fig. 1a). This analysis and consideration of previous VHL-targeting PROTACs pointed to the methyl group of the left-hand site (LHS) terminal acetyl group of VH032 as a suitable point of connection for a linker[35, 36]. A second solvent-exposed position available for derivatization was the phenyl group on the right-hand side (RHS), as previously employed with PROTACs targeting the Halotag[50]. To investigate the impact of derivatization, we designed three classes of Homo-PROTACS: (a) symmetric via the LHS acetyl group of each ligand (Fig. 1c); (b) symmetric via the RHS phenyl group (Fig. 1d); and (c) asymmetric via the acetyl group in one warhead and the phenyl in the other (Fig. 1e). In the cases b and c, at the underivatized terminal LHS we decided to retain either an acetyl (as in VH032) or a cyano-cyclopropyl moiety (as in VH298), a modification that led to increased binding affinities, cell permeability and cellular activities in the context of the VHL inhibitor alone[19]. To evaluate the potential impact of linker length, linkers comprised of polyethylene glycol chains with either three, four or five ethylene glycol units were chosen to connect the two VHL ligands.

It is known that the *trans* epimer of Hyp is an absolute requirement for VHL binding, and that the corresponding *cis* epimer abrogates binding to VHL, both within the context of a native HIF substrate peptide[51], and VHL ligands[19, 36]. We therefore designed two different PROTACs based on the structure of the first series (Fig. 1c), with the aim to use them as controls: a *cis-cis* epimer, expected to be completely inactive, and a *cis-trans* epimer compound, expected to retain binding to a single VHL

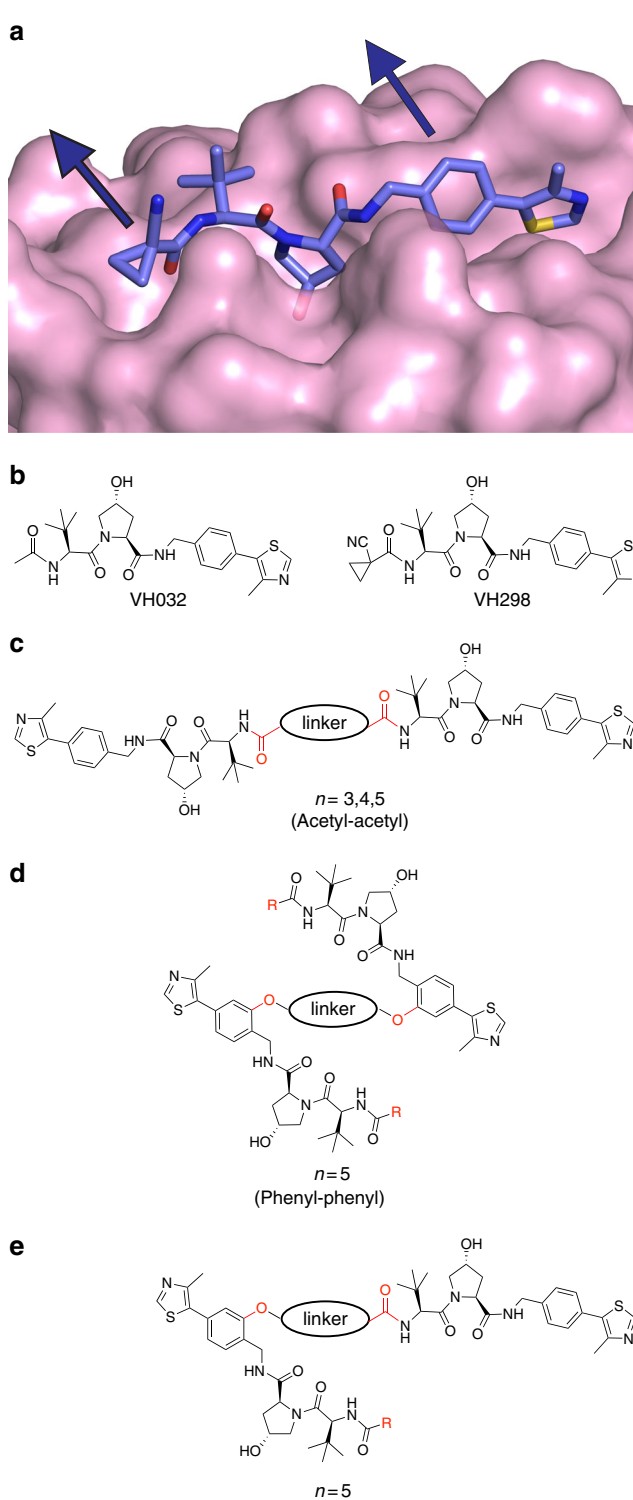

**Fig. 1** Structure-guided design of Homo-PROTACs to induce VHL dimerization. **a** Crystal structures of VHL in complex with VH298 (PDB code 5LLI)[19]. VHL is shown as a pink surface and the bound ligand as sticks representation with purple carbons, nitrogen atoms in blue, oxygen in red and sulphur in dark yellow. **b** Chemical structure of VHL inhibitors VH032 and VH298. **c–e** General chemical structures and design of Homo-PROTACs compounds. Linkage sites at the acetyl and phenyl groups are indicated in red

molecule in a 1:1 fashion, thus potentially acting as inhibitor but not as degrader.

**Synthesis**. For the synthesis of the first class of Homo-PROTACs (Fig. 1c), symmetric PEG linkers **4**, **5** and **6** bearing free carboxylate groups at either ends were obtained by reaction of *tert*-butyl bromoacetate with tri-, tetra- and penta-ethylene glycol in the presence of NaH in dioxane and followed, after purification, by treatment with 50% TFA in DCM (Fig. 2). The final compounds CM9, CM10 and CM11 were obtained by amide coupling of the VHL ligand **7** (prepared as previously described)[35] with linkers **4**, **5** and **6**, in a 2:1 ratio, respectively, in the presence of HATU as the coupling agent and DIPEA as the base (Fig. 2). For the synthesis of the symmetric *cis-cis* compound CMP98, compound **8**[35] was coupled with linker **6** to afford the desired product (Fig. 2). For the preparation of the asymmetric *cis-trans* compound CMP99, a synthetic route toward the synthesis of the monoprotected di-carboxylate linker was established. Pentaethylene glycol was the linker of choice because of ease of purification compared to longer PEGs, and at the same time yielding a control compound of average linker length (PEG-4 in this case). Pentaethylene glycol was converted in monobenzyl ether **9** in 71% yield, which was reacted with *tert*-butyl bromoacetic acid under biphasic conditions (DCM / 37% aq. NaOH and stoichiometric tetrabutyl ammonium bromide). After deprotection of the benzyl group by catalytic hydrogenation, formation of the carboxylic acid moiety was achieved by oxidation with TEMPO and bis-acetoxy iodobenzene (BAIB), delivering compound **11** in 65% yield (Fig. 3). Compound **7** was then coupled with linker **11** using the condition described above, affording compound **12**. Deprotection of the *tert*-butyl group using TFA and subsequently coupling with **8** afforded CMP99 in 66% yield (Fig. 3).

For the synthesis of the second class of symmetric Homo-PROTACs (Fig. 1d), it was decided to utilize compounds **17** and **18** as VHL warheads. Common precursor **16** was synthesized following a previously reported procedure[50], with minor modification that led to yield and purity improvements (see Supplementary Notes 1–3). Indeed, we observed that the use of HATU in combination with HOAT for the coupling steps of both Boc-*L*-Hyp and Boc-*tert*-leucine led to the formation of only the desired products, avoiding the formation of a bis-acylate secondary product[50], instead prominent when HATU was used alone. Compound **17** or **18** were obtained by treatment of compound **16** with 1-cyanocyclopropanecarboxylic acid in presence of HATU, HOAT and DIPEA or acetylimidazole and TEA (see Supplementary Methods). Synthesis of **17** was also performed using acetic anhydride, but during this reaction it was observed the formation of a secondary product di-acetylated, not only at the desired position but also at the hydroxyl group of the phenyl ring, which could however be separated. The PEG linkers for this class of compound were designed to contain a methanesulfonate group at either end, which could be coupled in a single step with the phenol of the VHL ligand. Linker **19** was prepared by mesylation of pentaethylene glycol and reacted with either compounds **17** or **18** in a 1:2 ratio in the presence of $K_2CO_3$ to afford CMP106 and CMP108, respectively, in good yield (Scheme 4). For the synthesis of asymmetric Homo-PROTACs, PEG **10** was converted in to the mesylated derivative **20** and reacted with **17** or **18** to obtain **21** and **22**, respectively in good yield (see Supplementary Notes 1–3). Final compounds CMP112 and CMP113 were obtained in good yield upon deprotection of the *tert*-butyl group and amide coupling with compound **7** (see Supplementary Notes 1–3).

**Biological evaluation**. We next tested all our Homo-PROTACs by monitoring protein levels after 10 h of compound treatment at 1 µM concentration in HeLa cells (Fig. 4a). We observed the

**Fig. 2** Synthesis of symmetric homo-PROTAC compounds derivatized from the terminal acetyl group. The shown route yielded symmetric *trans-trans* CM09, CM10 and CM11 and negative control symmetric *cis-cis* compound CMP98

striking effectiveness of CM09, CM10 and CM11 in inducing VHL depletion in cells (Fig. 4a), and a remarkably selective degradation for the closely migrating bands corresponding to the long isoform of VHL[52, 53], preferentially over the short isoform. The *VHL* gene includes three exons and it encodes two major isoforms of VHL: a 213 amino-acid long, 30 kDa form (pVHL30) and a 160 amino-acid long, 19 kDa form (pVHL19). pVHL19 lacks a 53 amino-acid long amino-terminal domain or N-terminal tail (pVHL-N), which is instead present in pVHL30. Although both isoforms are expressed in human cells, pVHL19 is the more prominent form in human tissues[54]. The most active compounds are symmetrically linked from the terminal LHS acetyl group of VH032. Linkage at different positions proved ineffective, suggesting a critical role played by the linking pattern. Control compounds CMP98 and CMP99 were unable to induce degradation of VHL (Fig. 4a), demonstrating that Homo-PROTAC activity is dependent on productive bivalent recruitment of VHL by the *trans* epimer. The length of the linker also seemed to affect cellular potency. Indeed, a decrease in effectiveness was observed at shorter linker lengths, with CM10 and CM11 being the most active compounds achieving total knockdown of pVHL30, followed by CM09 depleting 82% of the target protein. Interestingly, some degradation of the short isoform pVHL19 was also observed, albeit low (around 10% depletion). Levels of Cullin2, the central subunit of the CRL2-VHL complex[55], were also reduced upon treatment with CM10 and CM11 by up to 22% (Fig. 4a). Treatments with CM10 and CM11 also showed detectable albeit low increase in protein levels of the hydroxylated form of HIF-1α (Hdy-HIF-1α, Fig. 4a). As the parent inhibitor VH032 is completely ineffective at the same concentration of 1 µM (see ref. [19] and vide infra, Fig. 5), this effect cannot be due to VHL inhibition and is thought to be the result of compound-induced protein degradation. Levels of HIF-1α were, however, significantly lower than observed when VH032 was used at concentrations > 100 µM (Fig. 4a, see also ref. [19]). VHL knockdown by siRNA experiments in three different cell lines was consistent with CM11-induced knockdown, and confirmed that the bands observed indeed correspond to VHL (Fig. 4b).

It is known that VHL RNAi is insufficient to induce significant HIF stabilization (see also Fig. 4b) and does not cause detectable upregulation in HIF activity[56]. On the basis of these considerations, and the relatively low HIF stabilization observed with active Homo-PROTACs CM09–11 (Fig. 4a), it was presumed that compound treatment would not induce a HIF-dependent hypoxic response. To confirm this, we first used a luciferase reporter assay[57]. Hypoxia response element (HRE) -luciferase reporter HeLa-HRE and U2OS-HRE cells were treated with different concentrations of CM11 and at different times, and no increase in luciferase activity was detected relative to DMSO control treatment (Supplementary Fig. 1). These results were confirmed in a quantitative PCR with reverse transcription assay, where no upregulation of mRNA levels of the known HIF-target genes *CA9* was detected (Supplementary Fig. 2). Together the data suggest that un-degraded pVHL19 is sufficient to maintain low levels of HIF-1α.

We next turned our attention to further characterize the mode of action of the protein degradation induced by the active Homo-PROTACs CM09–11. To interrogate their relative cellular potency, dose-dependent treatments were performed at two different time points, 4 and 24 h before harvesting. All compounds confirmed preferential degradation of pVHL30 in a concentration-dependent manner, relative to the corresponding DMSO control (see Fig. 5a for CM11, and Supplementary Fig. 3 for CM09 and CM10). CM11 proved the most potent compound, inducing complete depletion of pVHL30 after 4 h already at 10 nM ($DC_{99} = 10$ nM, Fig. 5a). Selective pVHL30 knockdown was retained after 24 h, with half-degrading concentration ($DC_{50}$) between 10 and 100 nM. The effective degrading concentrations of CM11 are > 3 orders of magnitude lower than the inhibitory concentrations of the constitutive ligand VH032 alone, which is only active in cells at ~ 100 µM, underscoring the difference in cellular efficacy between the two mode of actions. Cellular levels of Cullin2 decreased by up to 73% upon treatment with CM11 (Fig. 5a). As previously observed, selective pVHL30 knockdown by Homo-PROTACs resulted in only minor increase in levels of HIF-1α, relative to hypoxia-inducing controls CoCl2, PHD

**Fig. 3** Synthesis of negative control homo-PROTAC compound derivatized from the terminal acetyl group. Compound CMP99 has *cis-trans* configuration

inhibitor IOX2 and VH032 (Fig. 5a). However, when tested at high micromolar concentrations, Homo-PROTACs acted preferentially as VHL inhibitors over VHL degraders, consistent with the so-called 'hook-effect' whereby formation of binary 1:1 complexes competes with and eventually supersedes the formation of the productive catalytic 2:1 complex[58]. Stabilization of Hdy-HIF-1α upon treatment with all three compounds at 100 μM was indeed comparable with the effect obtained with VH032 alone (Fig. 5a for CM11, and Supplementary Fig. 3 for CM09 and CM10). To confirm the cellular activities of Homo-PROTACs in a different cell line, U2OS cells were treated for 10 h with CM09, CM10 and CM11 using the same range of concentrations (1 nM–100 μM). A consistent profile of cellular activity was observed, confirming that the effects observed are independent from cell type (Supplementary Fig. 4).

We next interrogated the time-dependent activity of Homo-PROTACs. Progressive removal of VHL protein over time was observed, confirming selective depletion of pVHL30 over the short isoform (Fig. 5b for CM11 and Supplementary Fig. 5 for CM09 and CM10). In particular, CM11 was confirmed to be the most effective compounds, decreasing pVHL30 level by more that 70% already after 2 h of treatment, and essentially to completion after 8 h. The depletion effect was retained up to 12 h; however, interestingly, pVHL30 levels up to 11% were detected after 24–36 h treatment, to then decrease again after 48 h. Incomplete degradation of pVHL was observed upon treatment with CM09, even in the longer time points (Supplementary Fig. 5). The results obtained treating U2OS cells were consistent with what observed in the previous experiment. However, in this cell line all three compounds induced complete degradation of pVHL30 over time (Supplementary Fig. 4). We hypothesize that this could be due to the different expression

levels of VHL in the two cell lines. CM09 and CM10 achieved complete degradation of the target protein after 2 h of treatment. CM11 confirmed to be the most potent compound also in U2OS, achieving complete degradation of pVHL30 already after 1 h. Interestingly CM09 lost its cellular efficacy after 36 h. In contrast, both CM10 and CM11 retained their efficacy even at these longer time points (Supplementary Fig. 4).

To gain mechanistic insights in the mode of action of Homo-PROTACs, the dependency on CRL2-VHL and proteasome activities was examined. The reliance of the Homo-PROTAC-induced protein degradation on CRL2-VHL was assessed by inhibiting neddylation of Cullin2 using the NAE1 inhibitor MLN4924, which blocks the activity of CRLs, including CRL2-VHL[59]. Proteasome-dependency was interrogated by treating cells with the proteasome inhibitor MG132. To limit the known cytotoxicity of MLN4924 and MG132, HeLa cells where pre-treated with MLN4924 for 3 h followed by MG132 for 30 min before adding CM11 to the media, and cells were incubated for further 4 h before collecting. Single treatments with DMSO, MLN4924, MG132 and CM11 and combinations thereof were performed to disentangle the individual and combined effects of compound treatments. Degradation of pVHL30 induced by CM11 was completely abrogated when cells were pre-treated with MG132, establishing the expected proteasome-dependence of the chemical intervention (Fig. 6). CM11-induced degradation was also prevented by pre-treatment with MLN4924, confirming the dependency on the activity of CRL2[VHL] (Fig. 6). The same effect was observed when cells where co-treated with MLN4924 and MG132 before CM11 (Fig. 6). Immunoblots of Cullin2 levels confirmed the effective blockade of Cul2 neddylation by MLN4924 (Fig. 6). To assess if CM11 degrading activity was dependent on VHL binding, a competition experiment was

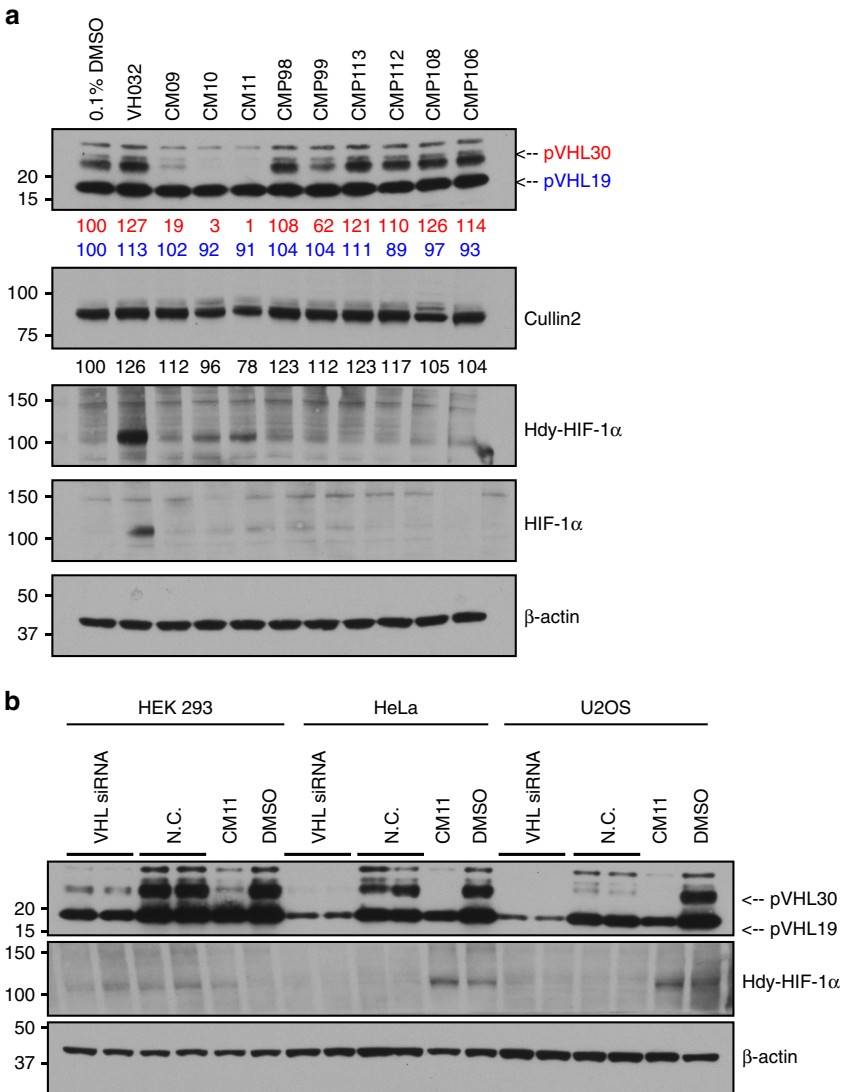

**Fig. 4** Homo-PROTACs CM09, CM10 and CM11 induce selective knockdown of the long isoform of VHL. **a** HeLa cells were treated with 0.1% DMSO, VH032 (150 μM) and 1 μM of the indicated compounds for 10 h. Abundance of individual proteins was analyzed by western blotting using corresponding specific antibodies accordingly after SDS-PAGE. The data demonstrated is a representative example of experiments performed in biological duplicates. **b** Different cells lines were treated with siRNA targeting VHL proteins or negative control siRNA (for 48 h), as well as with CM11 (1 μM) or 0.1% v/v DMSO for 10 h. The bands observed to selectively disappear in the presence of CM09–11 are closely migrating isoforms of pVHL30 (see refs [52, 53]). The data demonstrated is a representative example of experiments performed as four biological replicates

performed using the VHL inhibitor VH032[19]. HeLa cells were pre-treated with VH032 at 150 μM for 30 min before adding CM11 into the media. The plates were incubated for further 4 h before harvesting. As expected, VH032 blocked pVHL degradation (Fig. 6) consistent with the hypothesis that VHL induces degradation of itself. In contrast, pre-treatment with IOX4, a PHD2 inhibitor, did not impact CM11 activity (Fig. 6).

To evaluate the specificity of Homo-PROTAC-induced degradation, and identify potential off-targets, we performed isobaric tagging mass spectrometry proteomics to quantify degradation at the proteome level in an unbiased fashion. Amongst the 6,450 detected proteins that passed filtering criteria, no proteins other than Cul2 were substantially depleted by Homo-PROTACs CM09, CM10 or CM11 (1 μM for 10 h) compared to DMSO or VHL inhibitor treatment (Supplementary Data 1). Crucially, no effect on protein levels of other Cullins (Cul1, Cul3, Cul4A, Cul4B, Cul5 and Cul7) or CRL-associated subunits was observed (Supplementary Fig. 6). Together, the data demonstrate that

CM11 mainly induces depletion of pVHL30 and Cul2 but not other proteins. The proteomics data further evidenced no increase in protein levels of HIF-1α with Homo-PROTACs (but a small decrease with CM09) relative to DMSO (Supplementary Fig. 6). In contrast, treatment with 150 μM VH032 led to increased HIF-1α levels, as expected (Supplementary Fig. 6).

**Biophysical evaluation.** Key to the catalytic mode of action of PROTACs is the formation of a ternary complex[36, 38]. With Homo-PROTACs, VHL is presumed to act as both the E3 ligase and the substrate. Therefore, we next sought to monitor and biophysically characterize the ternary complex VHL:Homo-PROTAC:VHL that is thought to underlie cellular activity. To assess the formation of this ternary complex species in solution, isothermal titration calorimetry (ITC), size-exclusion chromatography (SEC) and AlphaLISA proximity assays were performed (Fig. 7). In ITC titration of CM11 against the VCB complex (VHL

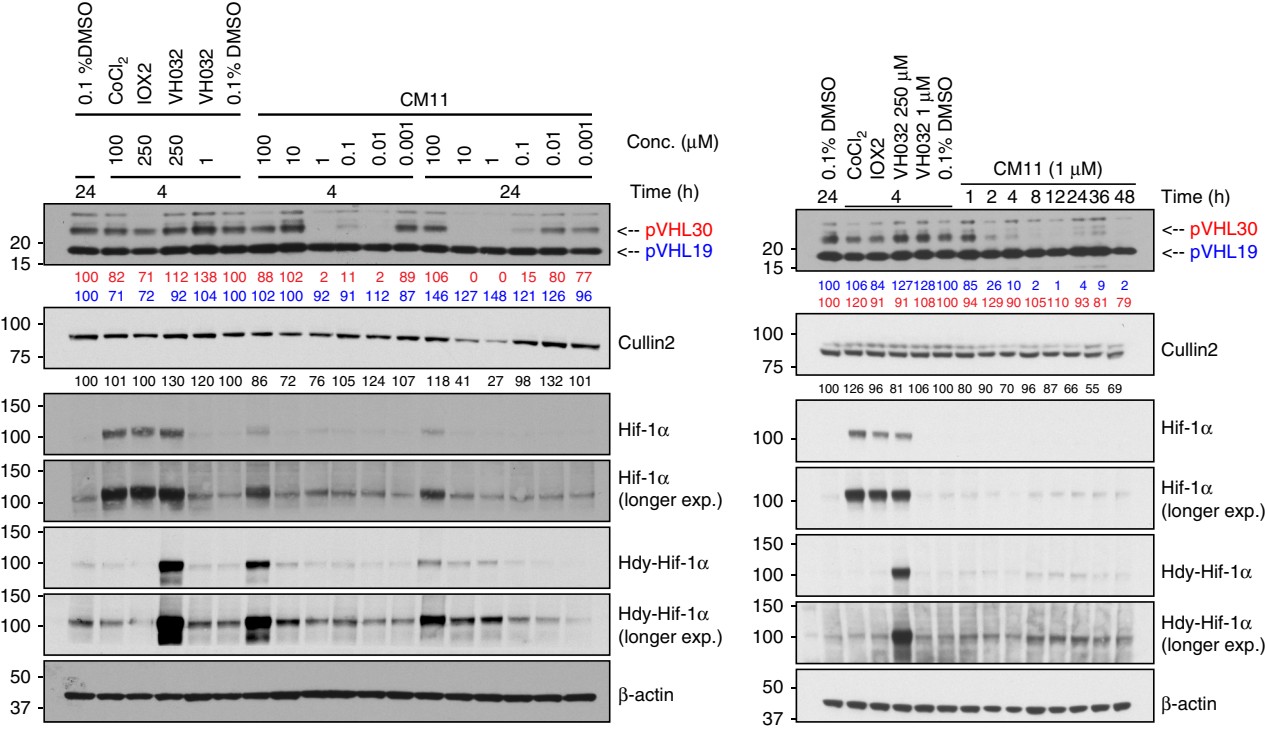

**Fig. 5** CM11 induces pVHL30 depletion in a concentration and time-dependent fashion. (left) Dose-response profile of HeLa cells treated with increasing concentration of CM11 for 4 or 24 h. (right) Time-course immunoblots of lysates from HeLa cells treated 1 μM of CM11 up to 48 h. Control treatments of 0.1% DMSO, CoCl₂ (100 μM), IOX2 (150 μM), and VH032 (250 μM or 1 μM) are included. All data demonstrated is a representative example of experiments performed in biological duplicates

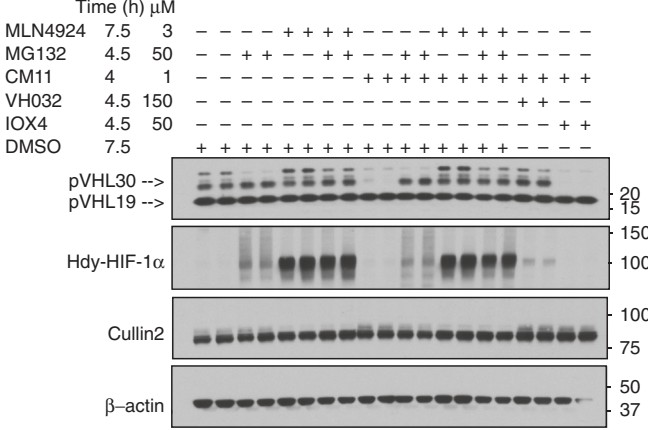

**Fig. 6** CM11 activity is CRL2$^{VHL}$ and proteasome-dependent. HeLa cells treated with CM11 in the absence or presence of proteasome inhibitor MG132, neddylation inhibitor MLN4924, VHL inhibitor VH032 or PHD2 inhibitor IOX4 as negative control. The data demonstrated here is representative of one biological replicate

with Elongin B and Elongin C)[60] the stoichiometry of binding ($n$ value) was found to be 0.6, compared to $n = 1$ with VH032 (Fig. 7a, Table 1). This result is consistent with CM11 binding to VHL in a 1:2 molar ratio, in contrast to VH032 that binds to VHL in a 1:1 ratio[18]. Notably, the $K_d$ value measured for CM11 was 11 nM (Table 1). Closer examination of the titration curve revealed that only one point features during the inflection of the curve. Indeed, because the protein concentration used in the experiment was 20 μM, the $c$ value (defined as $[P]_{tot}/K_d$) calculated for this experiment is 2500, which is above the upper limit of $c$ (~500–1000) that is a prerequisite for precise measurement of

binding affinity. Consequently, this analysis suggests that we may be underestimating the binding affinity of CM11, i.e., we can conclude that $K_d$ is ≤11 nM. This corresponds to an avidity (also known as cooperativity $\alpha$) of >18-fold when compared to VH032. Such large avidity of homo-bivalent molecules has been observed previously with other systems, for example the BET inhibitor MT1[48]. The binding interaction between CM11 and VHL was driven by a large apparent binding enthalpy ($\Delta H = -12.3$ kcal mol$^{-1}$), whereas the entropic term was slightly unfavorable ($-T\Delta S = 1.4$ kcal mol$^{-1}$). This observation underlines how the thermodynamic signature of CM11 is also very different when compared with that of VH032, in which case the binding $\Delta H$ was around half that observed with CM11, and both the enthalpic and entropic term contributed favorably to the $\Delta G$ of binding (Table 1). By contrast, the thermodynamic values obtained for CMP99 binding were entirely consistent with the ones of VH032 (Table 1). Specifically, CMP99 bound to VHL in a 1:1 ratio, as expected due to the presence of the *cis*-Hyp in one of the two moieties, and it exhibited comparable $\Delta H$ and $K_d$ values to VH032. As anticipated, binding was not detected with CMP98, the inactive *cis-cis* epimer. Superposition of integrated heat curves of CM11, CMP98 and CMP99 is shown in Fig. 7a and visually highlights the different behaviors of the three compounds. CM11 showed similar thermodynamic binding parameters relative to CM11, with $n$ value equal to 0.7 and a low $K_d$ of 32 nM. A stoichiometry close to 1 was instead found for CM09, suggesting that at the end of the titration this system was primarily populated by 1:1 complexes (Supplementary Figs 7 and 8), consistent with its lower avidity (Table 1)[58, 61]. To compare and contrast Homo-PROTACs binding to the long VHL isoform, we expressed and purified pVHL30 in complex with Elongin C and Elongin B (V₃₀CB) and performed ITC titrations (Table 1 and Supplementary Fig. 9). The thermodynamic parameters of binding of CMP99 to V₃₀CB were comparable to those to V₁₉CB, suggesting

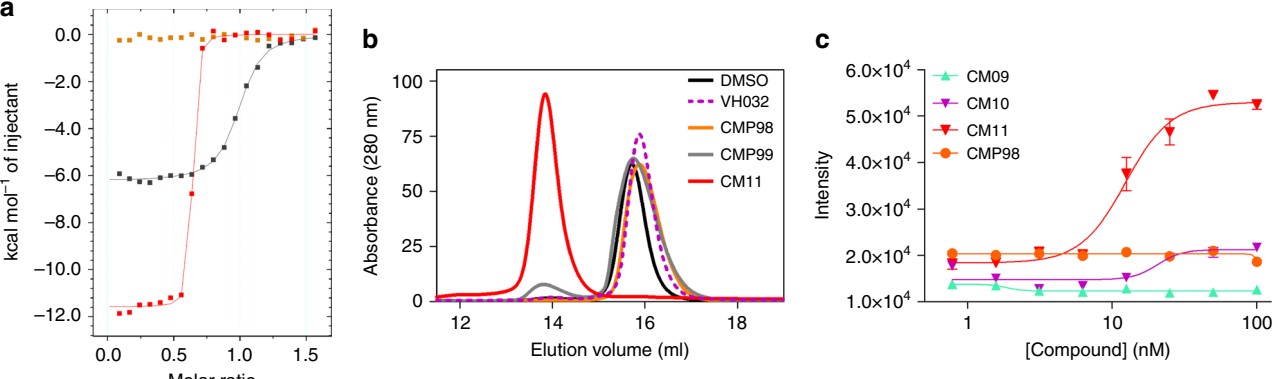

**Fig. 7** Biophysical studies of Homo-PROTACs binding to VHL. **a** Superposition of the integrated ITC heat curves of CM11 (blue), CMP99 (black) or CMP98 (green) titrations against VCB. **b** SEC assay of complex formation after incubation of CM11 (red), CMP98 (pale green), CMP99 (purple), VH032 (dotted purple) or DMSO (black) with VCB. **c** AlphaLISA: intensity values titrating CM09, CM10, CM11 and CMP98 against VCB. Each point is mean ($\pm$ SEM) intensity of four technical replicates

identical binary target engagement in vitro between the two isoforms. CM11 also exhibited large $\Delta H$ of binding and $n < 1$, albeit interestingly lower avidity for pVHL30 compared to pVHL19 (Table 1 and Supplementary Fig. 9).

SEC experiments showed that $V_{19}CB$ migrates more quickly in the presence of the active compound CM11 (2:1 protein:ligand ratio), relative to vehicle control (Fig. 7b). The shifted peak eluted at a volume corresponding to a species of ~ 90 kDa molecular weight, based on a calibration run with globular proteins of known molecular weight (see Methods), suggesting the peak corresponds to the ternary complex $(VCB)_2$:CM11. Formation of a 2:1 complex was also observed with $V_{30}CB$ (Supplementary Fig. 10). Mixing $V_{19}CB$, $V_{30}CB$ and CM11 resulted in a mixed population of the three species $(V_{19}CB)_2$:CM11, $(V_{19}CB)$:CM11:$(V_{30}CB)$ and $(V_{30}CB)_2$:CM11, suggesting that both VHL isoforms can form ternary complexes with CM11. No shift in VCB was observed following incubation with inactive CMP98, CMP99 or ligand VH032. Only in the sample containing CMP99 a small peak eluted at 13.5 ml (green curve, Fig. 7b). It is possible that such peak could be due to the formation of a lowly populated ternary complex. It is interesting that Schofield and colleagues observed weak binding of a *cis*-hydroxyprolyl containing HIF-1α peptide to VHL[51]. This weak binding, potentially enhanced by high avidity in the ternary complex, could be responsible for the small decrease of VHL levels observed during biological tests in cells (Fig. 3a). CM10 and CM09 showed formation of a ternary complex eluting at identical retention volume when compared to CM11 (Supplementary Fig. 11). No evidence of aggregation was seen with any of the compounds evaluated, as all observed peaks eluted well after the void volume.

Lastly, we employed an AlphaLISA proximity assay to compare ternary complex formation by CM09, CM10 and CM11. The assay showed the highest intensity signal for CM11, whereas negligible levels of complex formation were detected for CM09 and CM10 (Fig. 7c). Since SEC detected ternary species with all three compounds, the minimal intensity detected in the AlphaLISA likely reflects the inability of CM09 and CM10 to yield a significant ternary population at the low concentrations required for the assay. These results indicate that CM11 is the most effective Homo-PROTAC at driving ternary complex formation, consistent with CM11 exhibiting the highest avidity and full 2:1 stoichiometry in ITC. Together, the biophysical data supports CM11 as the most cooperative Homo-PROTAC in vitro, and provides a molecular rationale explaining its potent VHL-degrading activity inside cells.

## Discussion

We describe Homo-PROTACs, a small-molecule approach to effectively dimerize an E3 ubiquitin ligase to induce its own self-destruction. Using potent ligands for the E3 ligase VHL, we developed a series of homo-bivalent molecules that induce remarkably rapid, profound and selective degradation of the long isoform of pVHL at nanomolar concentrations. Compound-induced degradation was exquisitely dependent on the linkage pattern on the VHL ligand. The most active Homo-PROTAC, CM11, induces complete depletion of pVHL30 after 4 h already at 10 nM. Potent and selective degradation of pVHL30 was long-lasting, with a half-degrading concentration ($DC_{50}$) of <100 nM, a remarkable increase in cellular activity of >1000-fold compared to the parent inhibitor VH032. Mechanistically, we show that CM11 activity is strictly dependent on proteasome activity, Cul2 neddylation, and on VHL binding, and specifically on the formation of an avid 2:1 complex with VHL. Our data therefore supports a model in which a highly cooperative ternary complex $(VHL)_2$:CM11 functions as the key species responsible for the induced degradation of VHL itself (Fig. 8). Future structural studies of this ternary species are warranted. Interestingly, CM11 also led to a decrease in cellular levels of Cullin2, which we hypothesize to be the result of direct ubiquitination and degradation of Cullin2 as part of the $CRL2^{VHL}$ complex. To our knowledge, it is unprecedented that a PROTAC can induce the degradation of a protein forming part of the same complex with the protein targeted directly.

The preferential induced degradation of pVHL30 over the short VHL isoform was unanticipated and is an intriguing result of this work. This observation adds to recent evidence from us and others that chemical degraders designed from inhibitors recruiting more than a single protein can add a layer of target degradation selectivity independently of target engagement[35, 38, 41]. Biophysically, the VHL warhead was found not to distinguish between the two VHL isoforms at the level of binary target engagement. It was also found that CM11 does not considerably distinguish between pVHL19 and pVHL30 within ternary complexes. We therefore view it unlikely that the remarkable selectivity of VHL degradation is due to differences in molecular recognition. We also consider unlikely that preferential and more efficient lysine ubiquitination could play a role, because the extra region present in the long isoform (1–53) does not contain a single lysine residue. On the other hand, this region is predicted as intrinsically disordered[62], and it has been shown that proteins containing disordered N-terminal regions are more prone to

**Table 1 Thermodynamic binding parameters of complex formation of Homo-PROTACs measured by isothermal titration calorimetry**

| Protein | Compound | No. of replicates | $n$ | $K_d$ (nM) | $\alpha$ | $\Delta G$ (kcal mol$^{-1}$) | $\Delta H$ (kcal mol$^{-1}$) | $-T\Delta S$ (kcal mol$^{-1}$) |
|---|---|---|---|---|---|---|---|---|
| pVHL19 | VH032 (ref. [18]) | | $1.030 \pm 0.001$ | $188 \pm 6$ | – | $-9.17 \pm 0.02$ | $-5.53 \pm 0.01$ | $-3.65 \pm 0.02$ |
| | CM11 | 2 | $0.6 \pm 0.01$ | $11 \pm 2$ | 18 | $-10.9 \pm 0.1$ | $-12.3 \pm 0.7$ | $1.4 \pm 0.8$ |
| | CMP99[a] | 1 | $0.964 \pm 0.005$ | $146 \pm 2$ | – | $-9.33 \pm 0.06$ | $-6.23 \pm 0.05$ | $-3.1 \pm 0.7$ |
| | CM09 | 2 | $0.98 \pm 0.09$ | $41 \pm 15$ | 4 | $-10.3 \pm 0.2$ | $-6.9 \pm 0.3$ | $-3.5 \pm 0.5$ |
| | CM10 | 2 | $0.73 \pm 0.01$ | $32 \pm 5$ | 6 | $-10.2 \pm 0.1$ | $-9.4 \pm 0.1$ | $-0.8 \pm 0.2$ |
| pVHL30 | CM11[a] | 1 | $0.866 \pm 0.003$ | $25 \pm 3$ | 7.5 | $-10.4 \pm 0.1$ | $-11.3 \pm 0.1$ | $-0.9 \pm 0.1$ |
| | CMP99[a] | 1 | $1.050 \pm 0.004$ | $106 \pm 10$ | – | $-9.51 \pm 0.05$ | $-5.19 \pm 0.03$ | $-4.3 \pm 0.1$ |

All titrations were performed at 25 °C. Error values reported are the means ± 1 s.e.m., unless otherwise specified. Raw ITC data are shown for each titration in Fig. 7a and Supplementary Figs 7–9.
[a]Errors are generated by the Origin program and reflect the quality of the fit between the nonlinear least-squares curve and the experimental data.

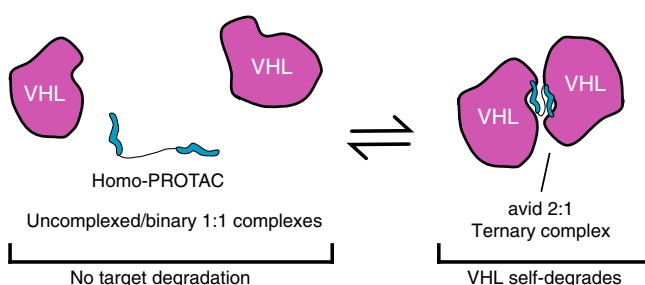

**Fig. 8** Proposed model for the mechanism of action of Homo-PROTAC CM11

proteasomal degradation[63]. It is also known that VHL is resistant to proteasomal degradation when in complex with ElonginB and ElonginC[64], so the form observed to be preferentially depleted may be free VHL, i.e., not bound to Elongins, or other proteasome-sensitive forms[65]. Previous work revealed different cellular localization patterns of pVHL19 and pVHL30, the former being primarily nuclear, while the latter predominantly cytoplasmic[56]. Crucially, pVHL30 was found to associate to microtubules, a function that is thought to be independent from its ligase activity[66]. We therefore speculate that the selective degradation of pVHL30 observed in cells could be the result of pVHL19 acting preferentially as part of CRL2-VHL, and pVHL30 as the 'neo-substrate'. Obtaining a more detailed biochemical and mechanistic understanding of the cellular mode of action of VHL Homo-PROTACs is of clear importance for future investigation.

Selective degradation of pVHL30 by CM11 led to minimal stabilization of HIF-α in cells, and as a result did not trigger HIF-dependent activity. This is consistent with complete knockdown of all VHL isoforms being required to achieve effective HIF stabilization in cells, as observed in $vhl^{-/-}$ cells such as VHL-deficient renal carcinoma cells[49]. Our results underscore the potential benefit of using CM11 to interrogate the biological function of specific VHL isoforms, without the masking downstream effects of a hypoxic response. Not much is known about the individual roles of VHL isoforms. Studies have highlighted how the 53-residue extra region of pVHL30 is not needed for tumor suppression[53], and how both isoforms can have HIF-dependent tumor suppressor functions in vivo[67, 68]. Other HIF-independent roles of pVHL have been proposed[69], including a role for pVHL in collagen assembly[70]. However, the individual roles of the different isoforms in these biological functions remain elusive. Moreover, many HIF-independent roles are thought to be independent upon Hyp recognition[71], and thus cannot be probed

chemically using the current Hyp-based VHL inhibitors[19]. Selective and acute knockdown of pVHL30 by CM11 therefore offers a new opportunity to address these questions.

In summary, we present CM11, a chemical probe for rapid and selective pVHL30 knockdown. CM11 provides a chemical tool alternative to conventional knockdown RNAi approaches and gene editing knockout technologies such as CRISPR-Cas9. Relevant information to the use of CM11 will be made available in the newly established 'Chemical Probes Portal' (http://www.chemicalprobes.org/)[72]. We anticipate CM11 will find wide use amongst chemical and cell biologists alike interested in investigating and dissecting the pleiotropic biological functions of pVHL. More generally, we provide proof-of-concept that bivalent molecules can be designed to induce an E3 ligase to destroy itself. This strategy opens powerful new avenues to drugging E3 ligases in ways that are not possible with inhibitors alone.

## Methods

**Synthesis of CM09, CM10, CM11 and controls CMP98 and CMP99**. *General method A*. PEG (1 eq.) was solubilised in dioxane anhydrous and NaH (2 eq.) was added under stirring. The resulting mixture was stirred at r.t. for 3h. The mixture was cooled down to 0 °C using ice bath and *tert*-butylbromo acetate (2 eq.) was added drop by drop. The resulting mixture was stirred at r.t O/N. The precipitate was filtered off and the organic phase evaporated to dryness. The resulting oil was taken up with ethyl acetate, washed with water, dried over MgSO$_4$ and evaporated to dryness. The resulting oil was purified by column chromatography using a gradient of ethyl acetate from 50% to 100% v/v in heptane.

*General method B*. tert-butyl esters **1**, **2**, **3** or **12** were dissolved in a solution of 50% v/v trifluoroacetic acid in DCM. The resulting solution was stirred for 1 h or until complete conversion of starting material. The solvent was removed under high vacuum. The resulting carboxylic acid was used as crude in the next step without any further purification. To a solution of carboxylic acid in 1 ml DMF were added HATU (1 eq.) and HOBT (1 eq.) and the pH of the reaction mixture was adjusted to > 9 by addition of DIPEA (3 eq.). The resulting solution was stirred at room temperature for 5 min and then amine **7** or **8** was added. The mixture was stirred at room temperature until no presence of the starting materials was detected by LC-MS. Water was added and the mixture was extracted with ethyl acetate (×3). The combined organic phases were washed with brine (×2), dried over MgSO$_4$ and evaporated under reduced pressure to give the corresponding crude, which was purified by HPLC using a gradient of 20% to 95% v/v acetonitrile in 0.1% aqueous solution of ammonia to yield the desired compound.

*di-tert-butyl 3, 6, 9, 12 – tetraoxatetradecanedioate* (1)

Following general method A, from triethylene glycol (1.125 g, 1 ml, 7.49 mmol, 1 eq.) in 10 ml of dioxane, NaH 60% in mineral oil (595.75 mg, 14.9 mmol, 2 eq.) and *tert*-Butyl bromoacetate (2.905 g, 2.19 ml, 14.9 mmol, 2 eq.), compound **1** was obtained as an oil after high vacuum. Yield: 538 mg, 1.42 mmol (19%). **¹H NMR**

(500 MHz, CDCl$_3$): δ 3.81 (s, 4H), 3.51–3.46 (m, 12H), 1.26 (s,18H). $^{13}$C NMR (126 MHz, CDCl$_3$): δ 169.1, 80.9, 70.1, 70.0, 68.5, 27.5. Analytical data matched those previously reported[73].

*di-tert-butyl 3,6,9,12,15-pentaoxaheptadecanedioate* (2)

Following general method A, from tetrathylene glycol (1.125 g, 1 ml, 5.49 mmol, 1 eq.) in 10 ml of dioxane, NaH 60% in mineral oil (463 mg, 11.5 mmol, 2 eq.) and *tert*-Butyl bromoacetate (2.25 g, 1.7 ml, 11.5 mmol, 2 eq.), compound **2** was obtained as an oil after high vacuum. Yield: 500 mg, 1.18 mmol (10%). $^1$H NMR (500 MHz, CDCl$_3$): δ 3.86 (s, 4H), 3.55–3.49 (m, 16H), 1.31 (s, 9H). Analytical data matched those previously reported[73].

*di-tert-butyl 3,6,9,12,15,18-hexaoxaicosanedioate* (3)

Following general method A, from pentaethylene glycol (1.126 g, 1 ml, 4.72 mmol, 1 eq.) in 10 ml of dioxane, NaH 60% in mineral oil (377 mg, 9.45 mmol, 2 eq.) and *tert*-Butyl bromoacetate (1.872 g, 1.7 ml, 11.5 mmol, 2 eq.), compound **3** was obtained as an oil after high vacuum. Yield: 300 mg, 0,641 mmol (14%). $^1$H NMR (400 MHz, CDCl$_3$): δ 3.94 (s, 4H), 3.66–3.56 (m, 20H), 1.40 (s, 18H). Analytical data matched those previously reported[73].

*N$^1$,N$^{14}$-bis((S)-1-((2S,4R)-4-hydroxy-2-((4-(4-methylthiazol-5-yl)benzyl)carbamoyl)pyrrolidin-1-yl)-3,3-dimethyl-1-oxobutan-2-yl)-3,6,9,12-tetraoxatetradecanediamide* (CM09)

Following general method B, from compound **1** (6.80 mg, 0.018 mmol, 1 eq.), compound **7** (20 mg, 0.045 mmol, 2.5 eq.), HATU (17 mg, 0.045 mmol, 2.5 eq), HOAT (6.12, 0.045 mmol, 2.5 mmol) and DIPEA (6.98 mg, 0.054 mmol, 3 eq) compound CM09 was obtained as a white solid. Yield: 8 mg, 0.007 mmol (40%). $^1$H NMR (400 MHz, CDCl$_3$): δ 8.61 (s, 2H), 7.48–7.45 (m, 2H), 7.31–7.27 (m, 8H), 7.23 (d, J = 10.2 Hz, 2H), 4.64–4.59 (m, 2H), 4.52–4.46 (m, 4H), 4.41–4.38 (m, 2H), 4.31–4.25 (m, 2H), 4.01–3.94 (m, 4H), 3.82 (d, J = 15.7 Hz, 2H), 3.62–3.52 (m, 12H), 2.45 (s, 6H), 2.42–2.34 (m, 2H), 2.12–2.06 (m, 2H), 1.19 (s, 2H), 0.89 (s, 18H); $^{13}$C NMR (101 MHz, CDCl$_3$): δ 170.2, 169.9, 169.6, 149.3, 147.5, 137.3, 130.6, 129.9, 128.4, 127.1, 69.9, 69.5, 69.3, 69.1, 57.6, 56.1, 55.9, 42.2, 35.5, 34.6, 25.4, 15.1. **HRMS (ESI)** m/z: [M + H]$^+$ calculated for: C$_{54}$H$_{74}$N$_8$O$_{12}$S$_2$: 1090.49; observed: 1091.5093.

*N$^1$,N$^{17}$-bis((S)-1-((2S,4R)-4-hydroxy-2-((4-(4-methylthiazol-5-yl)benzyl)carbamoyl)pyrrolidin-1-yl)-3,3-dimethyl-1-oxobutan-2-yl)-3,6,9,12,15-pentaoxaheptadecanediamide* (CM10)

Following general method B, from compound **2** (7.60 mg, 0.018 mmol, 1 eq.), compound **7** (20 mg, 0.045 mmol, 2.5 eq.), HATU (17 mg, 0.045 mmol, 2.5 eq), HOAT (6.12, 0.045 mmol, 2.5 mmol) and DIPEA (6.98 mg, 0.054 mmol, 3 eq) compound CM10 was obtained as a white solid. Yield: 6 mg, 0.005 mmol (30%). $^1$H NMR (500 MHz, CDCl$_3$): δ p.p.m., 7.58–7.55 (m, 2H), 7.35–7.30 (m, 10H), 4.72–4.67 (m, 2H), 4.57–4.47 (m, 6H), 4.38–4.33 (m, 2H), 4.15–3.91 (m, 8H), 3.68–3.56 (m, 20H), 2.51 (s, 6H), 2.45–2.38 (m, 2H), 2.20–2.14 (m, 2H), 0.95 (s, 18H). $^{13}$C NMR (126 MHz, CDCl$_3$): δ 170.3, 169.8, 169.6, 149.6, 146.9, 137.5, 129.5, 128.47, 128.40, 127.1, 70.0, 69.56, 69.52, 69.5, 69.2, 69.1,

57.7, 56.03, 55.98, 42.1, 35.6, 34.6, 25.4, 14.8. **HRMS (ESI)** m/z: [M + H]$^+$ calculated for: C$_{56}$H$_{78}$N$_8$O$_{13}$S$_2$: 1134.51; observed: 1135.5538.

*N$^1$,N$^{20}$-bis((S)-1-((2S,4R)-4-hydroxy-2-((4-(4-methylthiazol-5-yl)benzyl)carbamoyl)pyrrolidin-1-yl)-3,3-dimethyl-1-oxobutan-2-yl)-3,6,9,12,15,18-hexaoxaicosanediamide* (CM11)

Following general method B, from compound **3** (8.39 mg, 0.018 mmol, 1 eq.), compound **7** (20 mg, 0.045 mmol, 2.5 eq.), HATU (17 mg, 0.045 mmol, 2.5 eq), HOAT (6.12, 0.045 mmol, 2.5 mmol), DIPEA (6.98 mg, 0.054 mmol, 3 eq) compound CM11 was obtained as a white solid. Yield: 11.74 mg, 0.0099 mmol (55%). $^1$H NMR (400 MHz, CDCl$_3$): δ 8.61 (s, 2H), 7.41–7.38 (m, 2H), 7.29 (t, J = 7.6 Hz, 10H), 4.66–4.61 (m, 2H), 4.49–4.41 (m, 6H), 4.35–4.29 (m, 2H), 3.98–3.91 (m, 6H), 3.62–3.50 (m, 24H), 2.45 (s, 6H), 2.42–2.35 (m, 2H), 2.11–2.06 (m, 2H), 0.88 (s, 18H); $^{13}$C NMR (101 MHz, CDCl$_3$): δ 171.2, 170.9, 170.4, 150.3, 148.5, 138.3, 131.6, 130.9, 129.5, 128.1, 71.2, 70.61, 70.59, 70.5, 70.4, 70.3, 58.6, 57.0, 43.2, 36.5, 35.6, 26.4, 16.1. **HRMS (ESI)** m/z: [M + H]$^+$ calculated for: C$_{58}$H$_{82}$N$_8$O$_{14}$S$_2$: 1178.54; observed: 1179.6015.

*N$^1$,N$^{20}$-bis((S)-1-((2S,4S)-4-hydroxy-2-((4-(4-methylthiazol-5-yl)benzyl)carbamoyl)pyrrolidin-1-yl)-3,3-dimethyl-1-oxobutan-2-yl)-3,6,9,12,15,18-hexaoxaicosanediamide* (CMP98)

Following general method B, from compound **3** (7.12 mg, 0.028 mmol, 1 eq.), compound **8** (18.06, 0.040 mmol, 2.1 eq.), HATU (15.2 mg, 0.040 mmol, 2 eq.), HOAT (5.44 mg, 0.040 mmol, 2 eq.), DIPEA (7.45 mg, 0.0010 ml, 3 eq.), compound CMP98 was obtained as a white solid. Yield: 10.58 mg, 0.0089 mmol (45%). $^1$H NMR (400 MHz, CDCl$_3$): δ 9.09 (s, 2H), 8.02 (s, 2H), 7.31 (d, J = 8.5 Hz, 4H), 7.22 (d, J = 8.0 Hz, 4H), 7.16 (d, J = 9.2 Hz, 2H), 4.75–4.64 (m, 4H), 4.51 (d, J = 8.9 Hz, 2H,), 4.41–4.37 (m, 2H), 4.24–4.17 (m, 2H), 3.94 (d, J = 3.2 Hz, 4H), 3.84–3.81 (m, 4H), 3.62–3.54 (m, 20H), 2.49–2.47 (m, 2H), 2.44 (s, 6H), 2.26–2.17 (m, 4H), 0.93 (s, 18H); $^{13}$C NMR (101 MHz, CDCl$_3$): δ 173.2, 171.5, 169.7, 151.8, 138.8, 132.9, 129.5, 129.2, 128.3, 71.2, 71.1, 70.6, 70.48, 70.45, 70.4, 70.3, 59.9, 58.5, 56.5, 43.2, 35.6, 35.2, 26.4, 15.0. HRMS (ESI) m/z: [M + H]$^+$ calculated for: C$_{58}$H$_{82}$N$_8$O$_{14}$S$_2$: 1178.54; observed: 1179.6087.

*1-phenyl-2,5,8,11,14-pentaoxahexadecan-16-ol* (9)

Pentaethylene glycol (9.53 g, 50 mmol, 5 eq.) was added dropwise to a suspension of NaH 60% in mineral oil (800 mg, 20 mmol, 2.5 eq.) in 20 ml of DMF at 0 °C. The resulting mixture was stirred at r.t for 1 h. The reaction mixture was cooled to 0ºC, benzyl chloride (1 ml, 1.1 g, 8.72 mmol, 1 eq.) was added. The resulting mixture was stirred O/N at r.t. The reaction was quenched with a saturated solution of NH$_4$Cl and the aqueous phase was extracted with ethyl acetate (×3). The combined organic phases were dried over MgSO$_4$ and evaporated to dryness. The resulting oil was purified by column chromatography (from 0 to 60% of ethyl acetate in heptane) to afford the title compound as a oil. Yield: 2.055 g, 6.25 mmol (71%). $^1$H NMR (400 MHz, CDCl$_3$): δ 7.28–7.19 (m, 5H), 4.50 (s, 2H), 3.66–3.52 (m, 20H), 2.50 (s, 1H). $^{13}$C NMR (101 MHz, CDCl$_3$): δ 138.2, 128.3, 127.8, 127.6, 73.2, 72.7, 70.61, 70.58, 70.53, 70.51, 70.2, 69.4, 61.7

*tert-butyl 1-phenyl-2,5,8,11,14,17-hexaoxanonadecan-19-oate* (10)

To a stirred solution of **9** (2.055 g, 6.25 mmol, 1 eq.) in 12.8 ml of DCM was added 37% solution of NaOH (12.8 ml), followed by *tert*-butylbromo acetate (4.882 g, 25 mmol, 4 eq.) and TBABr (2118 mg, 6.37 mmol, 1.02 eq.). The resulting solution was stirred O/N at r.t. The reaction mixture was extracted with ethyl acetate (×3). The organic phases were combined and washed with brine (×1), dried over MgSO₄ and concentrate in vacuo. The resulting brow oil was purified by column chromatography (from 0 to 30% of ethyl acetate in petroleum) to afford the titled compound as colorless oil. Yield: 2.216 g, 5 mmol (80%). **¹H NMR** (500 MHz, CDCl₃): δ 7.28–7.20 (m, 5H), 4.50 (s, 2H), 3.95 (s, 2H), 3.65–3.55 (m, 20H), 1.40 (s, 9H). **¹³C NMR** (126 MHz, CDCl₃): δ169.7, 128.4, 127.7, 127.6, 81.5, 73.2, 70.7, 70.7, 70.6, 70.6, 69.4, 69.1, 28.1. **MS (ESI)** *m/z*: [(M-tBu) + H]⁺ calculated for: C₂₃H₃₈O₈: 442.26; observed: 387.20.

*19,19-dimethyl-17-oxo-3,6,9,12,15,18-hexaoxaicosanoic acid (11)*

**10** (2.216 g, 5 mmol, 1 eq.) was dissolved in 75 ml of ethanol, Pd/C (10 wt%) was added and the resulting mixture was place under hydrogen and stirred at r.t. until complete conversion of the starting material. The reaction mixture was filtered through celite, the celite pad was washed few times using ethanol. The filtrate was concentrated in vacuum to give an oil that was used for the next step without further purification. Yield: 1764 g, 5 mmol (quantitative). BAIB (3.546 g, 11.01 mmol, 2.2 eq.) and TEMPO (171.87 mg, 1.10 mmol, 0.22 eq.) were added to a solution of ACN/H₂O 1:1 containing previous obtained oil (1.764 g, 5 mmol, 1 eq.). The resulting mixture was stirred at r.t until complete conversion of the starting material. The crude was purified using ISOLUTE® PE-AX anion exchange column. The column was equilibrate with methanol, the reaction mixture poured in the column and let it adsorbed in the pad. The column was washed with methanol (×3) to elute all the unbound material. Then, the titled product was eluted using a 50% solution of formic acid in methanol. The organic phase was evaporated to dryness to afford the title compound as oil. Yield: 1.200 g, 3.27 mmol (65%). **¹H NMR** (400 MHz, CDCl₃): δ 4.12 (s, 2H), 3.98 (s, 2H), 3.72–3.60 (m, 16H), 1.43 (s, 9H). **¹³C NMR** (101 MHz, CDCl₃): δ 172.6, 169.7, 81.6, 71.0, 70.59, 70.56, 70.54, 70.46, 70.38, 70.35, 70.30, 68.9, 68.8, 28.1.

*tert-butyl (S)-19-((2S,4R)-4-hydroxy-2-((4-(4-methylthiazol-5-yl)benzyl)carbamoyl) pyrrolidine-1-carbonyl)-20,20-dimethyl-17-oxo-3,6,9,12,15-pentaoxa-18-azahenicosanoate (12)*

To a solution of PEG linker **11** (78.8 mg, 0.215 mmol, 1 eq.) in 1.5 ml DMF was added HATU (81.74 mg, 0.215 mmol, 1 eq.), HOAT (29.26 mg, 0.215 mmol, 1 eq.), DIPEA (80.13 mg, 0.106 ml, 0.645 mmol, 3 eq.) and the solution was stirred at room temperature for 5 min. Compound **7** (100 mg, 0.215 mmol, 1 eq.) was added and the pH of the reaction mixture was adjusted to > 9 by addition of DIPEA (80.13 mg, 0.106 ml, 0.645 mmol, 3 eq.). The mixture was stirred at room temperature until no presence of the starting materials was detected by LC-MS. The solvent was evaporated under reduced pressure to give the corresponding crude, which was purified by HPLC using a gradient of 20–95% v/v acetonitrile in 0.1% aqueous solution of ammonia to yield the titled compound as white solid. Yield: 75.6 mg, 0.094 mmol (44%). **¹H NMR** (400 MHz, CDCl₃): δ 9.00 (s, 1H), 7.45 (t, J = 5.9 Hz, 1H), 7.39–7.33 (m, 4H), 7.29 (d, *J* = 8.9 Hz, 1H), 4.71 (t, J = 8.0 Hz, 1H), 4.59–4.48 (m, 3H), 4.34 (dd, *J* = 5.2, 14.6 Hz, 1H), 4.08–3.92 (m, 5H), 3.69–3.61 (m, 18H), 2.52 (s, 3H), 2.47–2.41 (m, 1H), 2.19–2.11 (m, 1H), 1.46 (s, 9H), 0.97 (s, 9H). **¹³C NMR** (101 MHz, CDCl₃): δ 171.3, 171.1, 170.5, 170.0, 151.7, 139.1, 129.4, 128.3, 82.0, 71.1, 70.6, 70.4, 70.4, 70.3, 70.3, 70.2, 70.2, 68.9, 58.7, 57.3, 56.8, 43.1, 36.3, 35.1, 28.1, 26.4, 15.1. **MS (ESI)** *m/z*: [M + H]⁺ calculated for: C₃₈H₅₈N₄O₁₁S₂: 778.38; observed: 779.4.

*N¹-((R)-1-((2R,4R)-4-hydroxy-2-((4-(4-methylthiazol-5-yl)benzyl)carbamoyl) pyrrolidin-1-yl)-3,3-dimethyl-1-oxobutan-2-yl)-N¹⁷-((S)-1-((2S,4R)-4-hydroxy-2-*

*((4-(4-methylthiazol-5-yl)benzyl)carbamoyl)pyrrolidin-1-yl)-3,3-dimethyl-1-oxobutan-2-yl)-3,6,9,12,15-pentaoxaheptadecanediamide (CMP99)*

Following general method B, from compound **12** (75.6 mg, 0.094 mmol, 1 eq.) and trifluoroacetic acid (1 ml in 1 ml of DCM), the carboxylic acid derivative was obtained as oil. The crude was used for the next step without further purification. Yield: 70 mg, 0.094 mmol (quantitative). **MS (ESI)** m/z: [M + H]⁺ calculated for: C₃₄H₅₀N₄O₁₁S: 722.32; observed: 723.3. Following general method B, from compound **13** (5.5 mg, 0.006 mmol, 1 eq.), compound **8** (2.77 mg, 0.006 mmol, 1 eq.), HATU (2.28 mg, 0.006 mmol, 1 eq.), HOAT (1 mg, 0.006 mmol, 1 eq.), DIPEA (2.23 mg, 0.002 ml, 0.018 mmol, 3 eq.), **CMP99** was obtained as a white solid. Yield: 4.5 mg, 0.004 mmol (66%). **¹H NMR** (400 MHz, CDCl3): δ 8.74 (s, 1H), 8.73 (s, 1H), 7.37–7.34 (m, 9H), 7.18 (d, J = 8.9 Hz, 1H), 4.76–4.64 (m, 3H), 4.59–4.44 (m, 5H), 4.37–4.26 (m, 2H), 4.05–3.59 (m, 27H), 2.52 (s, 6H), 2.31–2.11 (m, 4H), 0.96 (s, 9H), 0.95 (s, 9H). **¹³C NMR** (101 MHz, CDCl₃): δ 173.0, 171.3, 170.0, 150.7, 145.4, 138.4, 129.53, 129.49, 128.2, 128.16, 71.2, 71.0, 70.54, 70.48, 70.4, 70.3, 58.4, 57.0, 56.7, 43.4, 35.2, 35.0, 26.4, 26.35, 15.9. **HRMS (ESI)** *m/z*: [M + H]⁺ calculated for: C₅₆H₇₈N₈O₁₃S₂: 1134.51; observed: 1135.5814.

**Biological and biophysical assays.** *Cell culture.* Human cell lines HeLa, U2OS and HEK 293, purchased from ATCC, were propagated in DMEM supplemented with 10% fetal bovine serum (FBS), L- glutamine, 100 µg ml⁻¹ of penicillin/streptomycin at 37 °C and 5% CO₂. Cells were maintained for no more than 30 passages. All cell lines were routinely tested for mycoplasma contamination using MycoAlert kit from Lonza. For compound treatment experiments, cells were transferred in six-well plates with either 3 × 10⁵ or 5 × 10⁵ cells per well in 2 ml of media. At 80% confluence, cells were treated with compounds at the desired concentration, reaching final DMSO concentration of 0.1% v/v. Cells were incubated at 37 °C and 5% CO₂ for the desired time before harvesting.

*Small interfering RNA.* For siRNA knockdown experiments, 3 × 10⁵ cells were seeded into each well of a six-well plate in order to achieve 70% of confluence on the day of transfection. siRNA (SMARTpool: ON-TARGETplus VHL siRNA L-003936-00-0005) was prepared as a 20 µM solution in RNase-free 1× siRNA buffer. Negative control siRNA (siRNA from Life Technologies, cat. # 4390843) was used as negative control. Medium was replaced on the day of transfection. siRNA solution (5 µL) of both VHL-targeting siRNA and negative control were added to 250 µl of Opti-mem in 1.5 ml tube. This solution was prepared in duplicate. Lipofectamine RNAiMax (5 µl) was added to 250 µl of Opti-mem in another 1.5 ml tube, also prepared in duplicate. The two solutions were combined and mixed by vortex and incubated at r.t. for 20 min. The whole volume of transfection mix was added to the six-well plate. Plates were incubated at 37 °C and 5% CO₂ for 48 h before harvesting.

*ML4924 and MG132 co-treatment.* Cells were transferred in six-well plates with 5 × 10⁵ cells per well in 2 ml media in order to achieve 80% confluence the day after. At t = 0, MLN4924 was added into the desired wells at 3 µM final concentration and 0.1% v/v of DMSO. DMSO (0.1% v/v final conc.) was added to the remaining wells in order to match identical conc. of vehicle in all wells. At t = 3 h, MG 132 was added into the desired wells at 50 µM final conc. and 0.1% v/v of DMSO. DMSO (0.1% v/v final conc.) was added to the remaining wells in order to achieve the same conc. of vehicle in all the wells. At t = 3.5 h, the desired wells were treated with 1 µM of CM11 in 0.1% v/v DMSO final concentration. DMSO (0.1% v/v final conc.) was added to the remaining wells to obtain the same conc. of vehicle in all the wells. The total final concentration of DMSO was therefore 0.3% v/v. Plates were incubated for 4 h at 37 °C and 5% CO₂ before harvesting. For competition experiments with VH032, cells were treated with VH032 at a final concentration of 150 µM (or IOX4 at 50 µM) for 30 min before treatment with CM11 at 1 µM final concentration for 4 h. Plates were incubated for the desired time at 37 °C and 5% CO₂ before harvesting.

*Immunoblotting.* Cells were lysed in lysis buffer (20 mM Tris pH 8, 150 mM NaCl, 1% Triton × 100) and a protease inhibitor cocktail (Roche) per 10 ml buffer. For protein extracts, the dishes were placed on ice. The media was aspirated and the tissue layer washed twice with ice-cold phosphate buffer saline (PBS). Lysis buffer (120 µl) was added and the cells detached from the surface with a cell scraper. After removal of the insoluble fraction by centrifugation, the protein concentration of the supernatant was determined by Pierce™ Coomassie (Bradford) Protein Assay Kit. Protein extracts were fractionated by SDS-PAGE on 4–12% Tris-Acetate NuPage® Novex® (Life Technologies) polyacrylamide gels and transferred to a nitrocellulose membrane using wet transfer. The membrane was then blocked with 5 % w/v Bovine serum albumin (BSA) in Tris-buffered saline (TBS) with 0.1% w/v Tween-20. For detecting proteins the following primary

antibodies in the given concentrations were used: anti-β-Actin (Cell Signaling Technology, 4970S, 13E5) 1:2000, anti-VHL (Cell Signaling Technology, #68547) 1:1000, anti-Hif-1α (BD Biosciences, 610959, clone 54) 1:1,000, anti-hydroxy-HIF-1α (Hyp564) (Cell Signaling Techonology; #3434) 1:1,000, anti-PHD2 (Bethyl Laboratories; A300-322A) 1:1,000, anti-PHD3 (Bethyl Laboratories; A300-327A) 1:1000, anti-CRBN (Proteintech; 11435-1-AP) 1:1,000. Following incubation with a horseradish peroxidase-conjugated secondary antibody (Cell Signaling Technology), the signal was developed using enhanced chemiluminescence (ECL) Western Blotting Detection Kit (Amersham) on Amersham Hyperfilm ECL film (Amersham). Band quantification was performed using ImageJ software and reported as relative amount as ratio of the each protein band relative to the lane's loading control. The values obtained were then normalized to 0.1% DMSO vehicle control. See Supplementary Fig. 12 for full-scanned images of western-blots shown in the main text figures.

*Luciferase assay.* Assay was performed as described[19]. Briefly, cells (HeLa and U2OS) stably expressing an HRE-luciferase reporter were treated for the indicated times with compounds. Cells were harvested in passive lysis buffer (Promega) and subjected to three freeze-thaw cycles. The soluble lysate fraction was used for assays, performed according to the manufacturer's instructions (Promega) using a Berthold Lumat LB 9507 Luminometer. Results were normalized for protein concentration, and reported as mean ± s.e.m. from three biological replicates.

*Quantitative real-time PCR.* Assay was performed as described[19]. Briefly, RNA was extracted from HeLa cell lysates using the RNeasy Mini Kit (Qiagen) and reverse transcribed using the iScript cDNA Synthesis kit (Bio-Rad). Real-time PCR was performed using PerfeCTa SYBR Green FastMix (Quanta Biosciences) in C1000 Touch Thermal Cycler (Bio-Rad). mRNA levels were calculated based on averaged Ct values from two technical replicates, normalized to mRNA levels of β-actin, and reported as mean ± s.e.m. from three biological replicates.

*MS proteomics. Sample preparation.* HeLa cells were seeded at $5 \times 10^6$ on a 10 cm plate 16 h before treatment. Cells were treated with 0.1% DMSO as vehicle control, VH032 (150 μM) to control for VHL inhibition, and CM09, CM10 or CM11 (1 μM), reaching a final DMSO concentration of 0.1% v/v. Cells were incubated at 37 °C and 5% $CO_2$ for 10 h before harvesting. Cells were placed on ice and medium was removed and cells were washed twice with cold PBS. Cells were lysed in 0.5 ml of 100 mM Tris pH 8.0, 4% (w/v) SDS supplemented with protease inhibitor cocktail (Roche). The lysate was pulse sonicated briefly and then centrifuged at $17,000 \times g$ for 15 min at 4 °C. The supernatant fraction of cell extract was snap-frozen in liquid nitrogen and stored in −80 °C freezer. Further sample processing, digestion, desalting, labeling and fractionation were performed as previously described[38].

*nLC-MS/MS analysis.* The TMT 10plex fractions were first analyzed as previously described[38]. Briefly, the TMT 10plex fractions were analyzed sequentially on a Q Exactive HF Hybrid Quadrupole-Orbitrap Mass Spectrometer (Thermo Scientific) coupled to an UltiMate 3000 RSLCnano UHPLC system (Thermo Scientific) and EasySpray column (75 μm × 50 cm, PepMap RSLC C18 column, 2 μm, 100 Å, Thermo Scientific). The MS2 isolation window was set to 0.4 Da and the resolution was set at 120,000. MS2 resolution was set at 60,000. The AGC targets for MS1 and MS2 were set at $3e^6$ ions and $1e5$ ions, respectively. The normalized collision energy was set at 32%. The maximum ion injection times for MS1 and MS2 were set at 50 and 200 ms, respectively. The dynamic exclusion time was set to 45 s. After the first analysis, the TMT labeled fractions were re-analyzed on the Q-Exactive HF using the same instrument settings and a target list generated from a theoretical trypsin digest of VHL. In this case only precursor ions that matched peptide ions from the target list were selected for ms/ms. The dynamic exclusion time was set to 30 s. After the second analysis, two fractions were further analyzed using a targeted SIM/ddMS2 method on the Q-Exactive HF. The targeted list for this run was generated from previous theoretical TMT labeled VHL protein digest. The methods were as follows, one SIM scan is followed by five targeted dd-MS2 scans. The SIM and MS2 isolation windows were set to 6 Da and 0.4 Da, respectively and the SIM scan and MS2 resolution set at 120,000 and 60,000 respectively. The AGC targets for MS1 and MS2 were set at $1e^5$ ions and $2e^5$ ions, respectively. The normalized collision energy was set at 32%. The maximum ion injection times for MS1 and MS2 were set at 200 and 300 ms, respectively. The dynamic exclusion time was set to 3 s.

*Peptide and protein identification.* All the raw MS data files from the three analyses were merged and searched against the Uniprot-sprot-Trembl-Human-Canonical database by Maxquant software 1.5.3.30 for protein identification and TMT reporter ion quantitation. The Maxquant parameters were set as follows: enzyme used Trypsin/P; maximum number of missed cleavages equal to two; precursor mass tolerance equal to 10 p.p.m.; fragment mass tolerance equal to 20 p.p.m.; variable modifications: Oxidation (M), Acetyl (N-term); fixed modifications: Carbamidomethyl (C). The data was filtered by applying a 1% false discovery rate followed by exclusion of proteins with less than two unique peptides. Quantified proteins were filtered if the absolute fold-change difference between the two DMSO replicates was ≥ 1.5.

*Protein expression and purification.* Human proteins VHL (UniProt accession number: P40337), ElonginC (Q15369), and ElonginB (Q15370) were used for all protein expression. The short-isoform-containing VCB complex ($V_{19}CB$) was expressed, purified and stored as described previously[74]. For expression of the long-isoform-containing VCB complex ($V_{30}CB$), N-terminally His$_6$-tagged VHL (1–213), ElonginC (17–112) and ElonginB (1–104) were co-expressed in

*Escherichia coli* BL21(DE3) at 20 °C for 18 h using 0.5 mM isopropyl β-D-1-thiogalactopyranoside (IPTG). E. coli cells were resuspended in binding buffer (50 mM HEPES, pH 8.0, 500 mM NaCl, 10 mM imidazole and 5 mM β-mercaptoethanol), lysed using a pressure cell homogenizer (Stansted Fluid Power), and lysate clarified by centrifugation. The supernatant was loaded onto a HisTrap FF affinity column (GE Healthcare) equilibrated with binding buffer, and bound protein was eluted using an imidazole gradient. The His$_6$ tag was removed using thrombin protease (Sigma), and the untagged complex was dialyzed into binding buffer. $V_{30}CB$ was then loaded onto the HisTrap FF column a second time, allowing impurities to bind as the complex eluted without binding. The flow-through was dialyzed using a low-salt buffer (20 mM HEPES, pH 8.0, 25 mM NaCl and 1 mM DTT) and then loaded onto a Resource Q anion exchange column (GE Healthcare); bound protein was eluted across a linear NaCl gradient. $V_{30}CB$ was then additionally purified by size-exclusion chromatography using a Superdex-75 16/600 column (GE Healthcare) equilibrated with 20 mM HEPES, pH 7.5, 100 mM sodium chloride and 1 mM TCEP. All chromatography purification steps were performed using Äkta FPLC purification systems (GE Healthcare).

*Isothermal titration calorimetry (ITC).* Titrations were performed on an ITC200 micro-calorimeter (GE Healthcare). Homo-PROTACs (CM11, CMP98 or CMP99) were diluted from a 100 mM DMSO stock solution to 150 μM in a buffer containing 20 mM Bis-tris propane, 150 mM NaCl, 1 mM tris(2-carboxyethyl) phosphine (TCEP), pH 7.4. The final DMSO concentration was 0.15% v/v. VCB protein experiments were carried out in a buffer containing 20 mM Bis-tris propane, 150 mM NaCl, 1 mM TCEP, 0.15% v/v DMSO, pH 7.4. The titrations consisted of 19 injections of 2 μL compounds solution (150 μM, in the syringe) at a rate of 0.5 μl s$^{-1}$ at 120 s time intervals into the VCB protein solution (20 μM, in the cell). An initial injection of compound solution (0.4 μL) was made and discarded during data analysis. All experiments were performed at 25 °C, whilst stirring the syringe at 600 rpm. The data were fitted to a single binding site model to obtain the stoichiometry $n$, the dissociation constant $K_d$ and the enthalpy of binding $\Delta H$ using the Microcal LLC ITC200 Origin software provided by the manufacturer.

*Size-exclusion chromatography (SEC).* SEC experiments were carried out in a ÄKTA pure system (GE Healthcare) at room temperature. The oligomeric state of the VCB complex in solution was analyzed by gel filtration in a buffer containing 20 mM Bis-Tris (pH 7), 150 mM NaCl and 1 mM 1,4-dithiothreitol (DTT) using a Superdex 200 Increase 10/300 GL column (GE Healthcare) and globular proteins of known molecular weight (GE Healthcare, 28-4038-41/42). VCB protein (50 μM) was incubated with CM11 (30 μM), CMP98 (30 μM), CMP99 (30 μM), VH032 (30 μM) or DMSO (0.5 %) for 20 min at room temperature prior to injection. Sample volume for each injection was 200 μl, and the flow rate was 0.5 ml min$^{-1}$. Peak elution was monitored using ultraviolet absorbance at 280 nm.

*Biotinylation of VCB.* The VCB complex was mixed with EZ-link NHS-PEG$_4$-biotin (Thermo Scientific) in a 1:1 molar ratio and incubated at room temperature for 1 h. The reaction was quenched using 1 M Tris-HCl, pH 7.5, and unreacted NHS-biotin was removed with a PD-10 MiniTrap desalting column (GE Healthcare) equilibrated with 20 mM HEPES, pH 7.5, 150 mM NaCl and 1 mM DTT.

*AlphaLISA assay.* All assays were performed at room temperature in 384-well plates with a final assay volume of 25 μL per well; plates were sealed with transparent film between addition of reagents. All reagents were prepared as 5× stocks diluted in 50 mM HEPES, pH 7.5, 100 mM NaCl, 0.1% (w/v) bovine serum albumin and 0.02% (w/v) 3-[(cholamidopropyl)dimethylammonio]-1-propanesulfonate (CHAPS). Biotinylated VCB (20 nM final) and His$_6$-VCB (20 nM final) were incubated with a range of Homo-PROTAC concentrations (0.5–200 nM; three-in-five serial dilution) for 1 h. Anti-His acceptor beads (PerkinElmer, 10 μg ml$^{-1}$ final) were added and plates were incubated for another hour. Streptavidin-coated donor beads (PerkinElmer, 10 μg ml$^{-1}$ final) were added and plates were incubated for a final 1 h. Plates were read on a PHERAstar FS (BMG Labtech) using an optic module with an excitation wavelength of 680 nm and emission wavelength of 615 nm. Intensity values were plotted against PROTAC concentration on a log$_{10}$ scale.

**Data availability**. Data supporting the findings of this study are available within the article (and its Supplementary Information files) and from the corresponding author upon reasonable request. The mass spectrometry proteomics data have been deposited to the ProteomeXchange Consortium via the PRIDE partner repository with the dataset identifier PXD007594.

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

## Acknowledgements

Financial support for this work was provided by the European Research Council (ERC-2012-StG-311460 DrugE3CRLs Starting Grant to A.C.), the Italian Ministry of Education, University and Research (Miur, PhD Studentship to C.M.), the Medical Research Council (MRC, grant number MC_UU_12016/2 to D.R.A.), and the Wellcome Trust (Strategic Award 100476/Z/12/Z for biophysics and drug discovery to the Division of Biological Chemistry and Drug Discovery). S.R. is funded by a CRUK Senior Fellowship (C99667/A12918). We thank Hannah Tovell and Dr Athanasios Karapetsas (MRC-PPU) for assistance with functional assays and Julianty Frost (BCDD/GRE) for discussion.

## Author contributions

A.C. supervised the project. C.M., S.J.H., A.T. and A.C. designed experiments. C.M. performed chemical synthesis, biological evaluation and ITC experiments. S.J.H. expressed and purified proteins, and performed SEC and AlphaLISA experiments; W.C. and D.J.L. performed MS proteomics experiments. C.M., S.J.H., A.T., S.R., D.R.A. and A.C. analyzed the data. S.R. and D.R.A. contributed reagents and tools; C.M., S.J.H. and A.C. wrote the manuscript with input from all authors.

## Additional information

**Competing interests:** The authors declare no competing financial interests.

