## [Peer Review File · Nature Communications]

Reviewers' comments:

Reviewer #1 (Remarks to the Author):

In this study, the authors reported the design, synthesis and evaluation of bivalent small-molecule dimerizers of the VHL E3 ubiquitin ligase to induce self-degradation. The novelty of this study is that although several classes of PROTACs have been successfully designed for different protein targets, this is the first of Homo-PROTACs. The study is well designed and the data support the conclusions. This manuscript is therefore suitable for publication in Nature Communications. The following revision is suggested.

In other PROTAC molecules, mass-spec experiments were often performed to demonstrate that the designed PROTAC molecules only degrade the intended target protein(s) but not other proteins. Although this is unlikely that the HOMO-PROTACs of VHL-1 degrade other proteins, it will be useful to perform a mass-spec analysis to be sure.

Reviewer #2 (Remarks to the Author):

Maniaci and colleagues report a novel chemical probe for selective Von Hippel Lindau (VHL) E3 ligase knockdown. E3s are the most diverse components in the ubiquitin proteasome system (UPS) and ubiquitylation affects most aspects of cellular life. Hence, defects in the UPS can cause cancer and many other diseases. However, therapeutic approaches targeting E3s in the ubiquitin pathways still lag behind. The absence of active sites suitable for drug targeting requires the usage of protein-protein interaction sites. Conventional inhibitors bind to the E3 substrate recognition site, however this competitive inhibition can pose limitations. High inhibitor concentrations are often required, which may result in pleiotropic effects. Recently described proteolysis targeting chimeric molecules (PROTACs) recruit selective proteins to an E3 ligase leading to the ubiquitylation and degradation of the non-native neo substrates. Here, the authors design homo-bivalent degrader molecules (Homo-PROTACs) that offer a novel tool to chemical knockdown of E3 ligases. Those chemical inducers of dimerization (CID) create complexes in which the targeted E3 ligase acts as the enzyme and the substrate at the same time leading to an efficient and specific self degradation. As an example the authors target the well-studied von Hippel-Lindau (VHL) E3 ligase. By the design of diverse Homo-PROTACs they characterize the most efficient compound, which can induce a rapid and isoform-specific self degradation of the ligase.

The paper is well written and experiments are well designed. The compounds may have limited use as experimental tools, but the concept of CID compounds is very interesting and could be expanded to other E3 ligases, and one could even envision therapeutic use of these type of compounds in the future. The "catalytic" mode of action is clearly an advantage over more traditional inhibitory molecules. My main concern is that a more thorough characterization of on and off target effects should be performed because these compounds will be used by researchers as tools to inhibit VHL function. Unless we know about potential off target effects the use of these compounds may lead to misinterpretation of data in future studies. For example the authors could compare expression profile changes induced by VHL knockdown and compound treatment to evaluate potential off target effects. I feel this is an important point to avoid future misinterpretation of results. Even though the "catalytic" mode of action allows relatively low compound concentrations, the main advantage over traditional knockdown experiments is the fast action of tool compounds. To achieve fast action used compound concentrations are in fact quite high. I envision that most applications will use these compounds to achieve VHL inhibition within a few hours. It is therefore important to characterize off target effects experimentally keeping these applications in mind (i.e. 2h, 1uM C11). If the authors can address this

concern I support publication in Nature Communications.

minor concerns

1) Fig. 5 left: band intensities for 100 and 10 μ M C11 (VHL30) do not match visual impression. In general, some of the numbers seem inconsistent with the visual inspection (lanes 5 and 6 VHL30, same blot, and several others). The authors should recheck the quantitation of band intensities throughout the manuscript.

2) In general I feel some of the key experiments would benefit from statistical analysis of biological replicates.

Manuscript: "Homo-PROTACs: bivalent small-molecule dimerizers of the VHL E3 ubiquitin ligase to induce self-degradation" (Manuscript ID#: NCOMMS-17-09302)

We thank the Editors and Referees for their critical reading of our manuscript and for the insightful and constructive suggestions on how to improve the study. We are encouraged by the overall supporting and enthusiastic reviews, and have used their input to improve our manuscript. We have endeavored to respond experimentally to the main suggestion concerning characterization of potential off-target induced degradation by homo-PROTACs.

Below are our detailed responses (in red) to each of the Referees' specific points, and we believe to have sufficiently addressed their suggestions and concerns, resulting in a revised manuscript that is much improved from the review.

Reviewer #1

Remarks to the Author:

In this study, the authors reported the design, synthesis and evaluation of bivalent small-molecule dimerizers of the VHL E3 ubiquitin ligase to induce self-degradation. The novelty of this study is that although several classes of PROTACs have been successfully designed for different protein targets, this is the first of Homo-PROTACs. The study is well designed and the data support the conclusions. This manuscript is therefore suitable for publication in Nature Communications. The following revision is suggested.

Thank you for the strong vote of support!

In other PROTAC molecules, mass-spec experiments were often performed to demonstrate that the designed PROTAC molecules only degrade the intended target protein(s) but not other proteins. Although this is unlikely that the HOMO-PROTACs of VHL-1 degrade other proteins, it will be useful to perform a mass-spec analysis to be sure.

We thank the Referee for this suggestion. We agree with the Referee that a general assessment for off-target effects of a new chemical probe is worthwhile. As advised, we have performed TMT-labeling MS proteomics experiments to characterize potential off-target effects in an unbiased fashion at the global proteome level. The results are now included in the revised manuscript (**Supplementary Data 1**). The data evidenced no substantial depletion of proteins other than the on-target proteins (pVHL30 and Cul2) upon treatment with homo-PROTACs CM09, CM10 and CM11 compared to DMSO vehicle control or VHL inhibition by the parent warhead ligand VH032. Crucially, no induced degradation of other Cullins or CRL-associated subunits other than Cul2 was observed (data shown in new **Supplementary Fig. 6**). These data support CM11 as a chemical probe for selective pVHL30 knockdown, in particular at effective doses between 100 nM to 1 μ M.

Reviewer #2

Remarks to the Author:

Maniaci and colleagues report a novel chemical probe for selective Von Hippel Lindau (VHL) E3 ligase knockdown. E3s are the most diverse components in the ubiquitin proteasome system (UPS) and ubiquitylation affects most aspects of cellular life. Hence, defects in the UPS can cause cancer and many other diseases. However, therapeutic approaches targeting E3s in the ubiquitin pathways still lag behind. The absence of active sites suitable for drug targeting requires the usage of protein-protein

interaction sites. Conventional inhibitors bind to the E3 substrate recognition site, however this competitive inhibition can pose limitations. High inhibitor concentrations are often required, which may result in pleiotropic effects. Recently described proteolysis targeting chimeric molecules (PROTACs) recruit selective proteins to an E3 ligase leading to the ubiquitinylation and degradation of the non-native neo substrates. Here, the authors design homo-bivalent degrader molecules (Homo-PROTACs) that offer a novel tool to chemical knockdown of E3 ligases. Those chemical inducers of dimerization (CID) create complexes in which the targeted E3 ligase acts as the enzyme and the substrate at the same time leading to an efficient and specific self degradation. As an example the authors target the well-studied von Hippel-Lindau (VHL) E3 ligase. By the design of diverse Homo-PROTACs they characterize the most efficient compound, which can induce a rapid and isoform-specific self degradation of the ligase.

The paper is well written and experiments are well designed. The compounds may have limited use as experimental tools, but the concept of CID compounds is very interesting and could be expanded to other E3 ligases, and one could even envision therapeutic use of these type of compounds in the future. The “catalytic” mode of action is clearly an advantage over more traditional inhibitory molecules.

We appreciate for the strong vote of support, thank you!

My main concern is that a more thorough characterization of on and off target effects should be performed because these compounds will be used by researchers as tools to inhibit VHL function. Unless we know about potential off target effects the use of these compounds may lead to misinterpretation of data in future studies. For example the authors could compare expression profile changes induced by VHL knockdown and compound treatment to evaluate potential off target effects. I feel this is an important point to avoid future misinterpretation of results. Even though the “catalytic” mode of action allows relatively low compound concentrations, the main advantage over traditional knockdown experiments is the fast action of tool compounds. To achieve fast action used compound concentrations are in fact quite high. I envision that most applications will use these compounds to achieve VHL inhibition within a few hours. It is therefore important to characterize off target effects experimentally keeping these applications in mind (i.e. 2h, 1µM C11). If the authors can address this concern I support publication in Nature Communications.

We thank the Reviewer for these positive and supportive comments. We agree with the Referee that a general assessment for potential off-target effects of a new probe is valuable. As advised, we have performed TMT-labeling MS proteomics experiments to characterize potential off-target effects in an unbiased fashion at the global proteome level. We treated cells with 1 µM homo-PROTACs CM09, CM10 and CM11 for 10 h, as this time-point achieved complete depletion of pVHL30 with 1 µM CM11 in our time-dependent experiment. Control treatments with DMSO and the warhead ligand VH032 were included. Experiments were performed in biological duplicates.

The data (now included in the revised manuscript as **Supplementary Data 1) evidenced no substantial depletion of proteins other than the on-target proteins (pVHL30 and Cul2) upon treatment with homo-PROTACs CM09, CM10 and CM11 compared to DMSO vehicle control or VHL inhibition. Crucially, no induced degradation of other Cullins or CRL-associated subunits other than Cul2 was observed (data shown in new **Supplementary Fig. 6**).**

The reviewer’s suggestion to compare gene expression changes between genetic *versus* compound induced VHL knockdown is interesting. We show that compound-induced VHL knockdown is selective for the long isoform, and does not completely deplete the short isoform. As a result compound treatment does not induce a HIF-dependent transcriptional response, which instead requires complete VHL knockdown or VHL inhibition, as we have previously shown (Frost et al., Nat Comm. 2016). Moreover, it has remained challenging to genetically induce selective knockdown of the pVHL long isoform, because expression of the short isoform is the result of an alternative translation start site (PMCID: PMC21160). Obtaining a more detailed biochemical understanding of the downstream

consequences of selective pVHL30 knockdown will be of clear importance for future investigation.

minor concerns

1) Fig. 5 left: band intensities for 100 and 10 uM C11 (VHL30) do not match visual impression. In general, some of the numbers seem inconsistent with the visual inspection (lanes 5 and 6 VHL30, same blot, and several others). The authors should recheck the quantitation of band intensities throughout the manuscript.

We regret the imprecision in the quantification of some of the numbers, thank you for spotting these. As advised, we have carefully rechecked the quantitation of band intensities throughout the manuscript, and revised them accordingly.

2) In general I feel some of the key experiments would benefit from statistical analysis of biological replicates.

We appreciate the comment. We now include p-value statistical analysis where appropriate. We have also included comments clarifying the number of biological replicates of which the western blot data presented are representative images.

REVIEWERS' COMMENTS:

Reviewer #1 (Remarks to the Author):

The authors have performed the requested experiments and the manuscript is now ready for publication.

Reviewer #2 (Remarks to the Author):

The authors addressed my concerns in the revised manuscript by Maniaci et al. The new MS data add further evidence for selectivity of the compound. The authors are correct that expression profiling experiments I suggested are not feasible approaches to address this question. Importantly the authors have strengthened the data by providing information about reproducibility of data and reevaluated quantitations. I therefore support publication of the manuscript in Nature communications .